# Evaluation of sample pooling using Xpert Carba-R and Xpert *vanA*/*vanB* PCR for screening of carbapenemase-producing *Enterobacterales* and vancomycin-resistant *Enterococcus* colonization

Danielle Keidar-Friedman,[1,2] Larissa Gil,[2] Anka Tsur,[2] Tal Brosh-Nissimov,[3,4] Yehuda Carmeli,[5,6] Boaz David Rosenfeld,[7] Nadav Sorek[1,2]

**ABSTRACT** Screening for carbapenemase-producing *Enterobacterales* (CPE) and vancomycin-resistant *Enterococcus* (VRE) colonization among hospitalized patients is a standard infection control procedure that also guides appropriate antibiotic treatment in healthcare settings. Extensive CPE screening in low-prevalence regions imposes a considerable laboratory workload and substantial costs that can be mitigated through the utilization of pool testing. In this study, we evaluated PCR pooling for the detection of CPE and VRE colonization from rectal swabs collected in our hospital, using the Xpert Carba-R and Xpert *vanA*/*vanB* assays. CPE pooling demonstrated excellent performance, with 94.44% sensitivity (74.2–99.0 CI 95%), 100% specificity (99.1–100), 100% positive predictive value (81.6–100), and 99.78% (98.7–100) negative predictive value (NPV). In contrast, VRE detection showed lower performance, with 75.86 (63.5–85.0 CI 95%) specificity and 100% (92.0–100) NPV, primarily due to false-positive results for the *vanB* gene. Overall, the pooling technique showed great potential for CPE screening in low-prevalence settings. Further studies are needed to evaluate VRE pooling and clarify the clinical utility of PCR results for *vanB*, which currently requires culture confirmation.

**IMPORTANCE** Carbapenemase-producing *Enterobacterales* (CPE) and vancomycin-resistant enterococci (VRE) pose a major threat to healthcare systems due to their role in hospital-acquired infections and limited treatment options. Routine screening for CPE/VRE colonization is essential for preventing transmission, yet conventional culture-based methods are labor-intensive and time-consuming. This study demonstrates that PCR pooling is a highly effective and resource-efficient approach for CPE screening in low-prevalence settings, maintaining high sensitivity and specificity while reducing laboratory workload. The performance of VRE pooling, particularly for *vanB* detection, requires further evaluation. These findings support the adoption of pooling strategies for CPE surveillance and highlight the need for improved molecular diagnostics for VRE, offering a promising solution to address growing challenges in clinical microbiology diagnostics.

**KEYWORDS** clinical microbiology diagnostics, vancomycin-resistant enterococci, carbapenemase-producing *Enterobacterales*, healthcare-associated infections, antimicrobial resistance

Carbapenem-producing *Enterobacterales* (CPE) and vancomycin-resistant enterococci (VRE) are a growing threat worldwide and have been proven to cause epidemic outbreaks in hospital settings, which can lead to increased mortality rates in hospitalized

**Peer Reviewers** Masahiro Suzuki, Fujita Ika Daigaku, Toyoake, Japan; Ya Wang, Harvard University, Allston, Massachusetts, USA

Address correspondence to Danielle Keidar-Friedman, danielleke@assuta.co.il.

B.D.R. is an employee of Medison Pharma Ltd, a representative of Cepheid diagnostic products in Israel. The other authors declare no competing interests.

patients (1, 2). Generally, resistance mechanisms in bacteria can be intrinsic or acquired by mobile genetic elements such as plasmids and integrons. While many different bacterial species harbor intrinsic mechanisms of resistance to some type of antibiotic, acquired resistance is much more challenging due to its potential spread between different bacterial species (3).

These organisms are mostly acquired during hospitalization in acute-care hospitals or long-term care facilities. The major site of carriage is the lower gastrointestinal tract (4, 5). Although colonization can be asymptomatic, affected patients can shed bacteria into the hospital environment and become a source of transmission to other patients. These organisms can colonize not only the gut but also the skin and respiratory tract, potentially leading to infections (6). Given the limited treatment options for infections caused by these multi-drug-resistant pathogens and their potential spread within healthcare settings, comprehensive infection prevention measures are essential. These include screening through rectal swab tests, isolation or cohorting of carriers, and thorough environmental cleaning and disinfection (4, 7).

Many healthcare facilities implement active surveillance programs to detect carriers among high-risk populations using rectal or perianal swab testing (8, 9). In the standard clinical microbiology procedure, these samples are usually plated on selective chromogenic media for an overnight incubation. Following the growth and identification of bacterial colonies, the presence of carbapenemase genes or enzymes is examined (4). Swabs for VRE screening are usually plated on chromogenic agar, incubated overnight, and then tested for vancomycin-resistance genes, *vanA* and *vanB* (10). The turnaround time using these culture-based methods ranges between 18 and 72 hours (4, 10).

Although culture-based methods are considered highly accurate and reliable, in recent years, there has been an increase in the use of molecular assays for CPE/VRE screening directly from samples, as these usually result in shorter turnaround time (TAT) and higher sensitivity (7). The Xpert Carba-R is a diagnostic PCR test for the rapid identification of common plasmid-encoded carbapenemases: blaKPC, blaOXA-48, blaNDM, blaVIM, and blaIMP using the GeneXpert platform. The use of the test is relatively easy for both professional and unprofessional staff, and the test can be completed within 1 hour. Studies that have evaluated the Xpert Carba-R test have reported very high detection rates and sensitivity values (11). Another advantage of gene detection methods is detecting these carbapenemase genes in non-*Enterobacterales* of interest, i.e., *Pseudomonas* spp. and *Acinetobacter baumannii*, which are often missed by culture-based methods directed to detect *Enterobacterales*. Good diagnostic results were also found for the use of Xpert *vanA/vanB*; however, the test was found to be more accurate in detecting enterococci carrying *vanA* than *vanB* (12). Nevertheless, these tests are considerably more expensive, and therefore, many laboratories still rely on culture-based methods.

The COVID-19 pandemic has demonstrated the utility of PCR pooling as a cost-effective testing strategy (13, 14). This approach, which combines multiple samples into a single test, could potentially be applied to other clinical diagnostic tests. When a pooled sample tests negative, all individual samples are considered negative, while positive pools require individual retesting of each sample. In this study, we have evaluated a rapid PCR pooling method for the detection of carbapenem and vancomycin resistance genes that can indicate the carriage of CPE or VRE among inpatients.

## MATERIALS AND METHODS

### Study design

The study was conducted in two phases: a pre-clinical phase and a clinical phase. In the pre-clinical phase, a retrospective evaluation was performed on previously tested rectal swab samples to evaluate the performance of molecular PCR testing on pooled swab specimens. In the clinical phase, prospectively collected rectal swab samples were tested using a pooling strategy and compared with the reference culture-based method,

which included PCR confirmation for suspected colonies (see Fig. 1). The clinical phase was conducted in March 2024 at Samson Assuta Ashdod University hospital—a tertiary hospital with 300 beds and a prevalence rate of 1.4% for CPE and 4.7% for VRE.

## Pre-clinical phase

### Cultures

Cotton swab samples collected for CPE screening (Copan Diagnostics, Murrieta, CA, USA) from high-risk inpatients were cultured on chromogenic selective agar plates (Supercarba Hylabs, Rehovot, Israel, or CHROMagar CPE, BD, NJ, USA) and incubated for 18–24 hours at 35°C ± 2°C. In cases of colony growth, colonies were tested by matrix-assisted laser desorption/ionization (MALDI-TOF MS, bioMérieux, Marcy-l'Étoile, France) for species identification. If an *Enterobacterales* sp. was identified, the isolate was tested by Xpert Carba-R PCR (Cepheid, Sunnyvale, CA, USA) on the GeneXpert platform (Cepheid, Sunnyvale, CS, USA) for the detection of carbapenemase genes. Detection of one or more carbapenemases was considered a positive result for CPE. Samples showing no carbapenemase genes or growth of non-*Enterobacterales* species were considered negative for CPE colonization. Cotton swab samples collected for VRE screening (Copan Diagnostics, Murrieta, CA, USA) were cultured on CHROMagar VRE (Hylabs, Rehovot, Israel) and incubated for 18–24 hours at 35°C ± 2°C. In cases of colony growth, colonies were tested by MALDI-TOF-MS for species identification. If *Enterococcus faecium/faecalis* was identified, it was further tested by Xpert *vanA/B* PCR for the detection of vancomycin resistance genes. Detection of *vanA/B* was considered a positive result for VRE. If no *vanA/B* genes were detected or the growth was from other species, the sample was considered negative for VRE colonization.

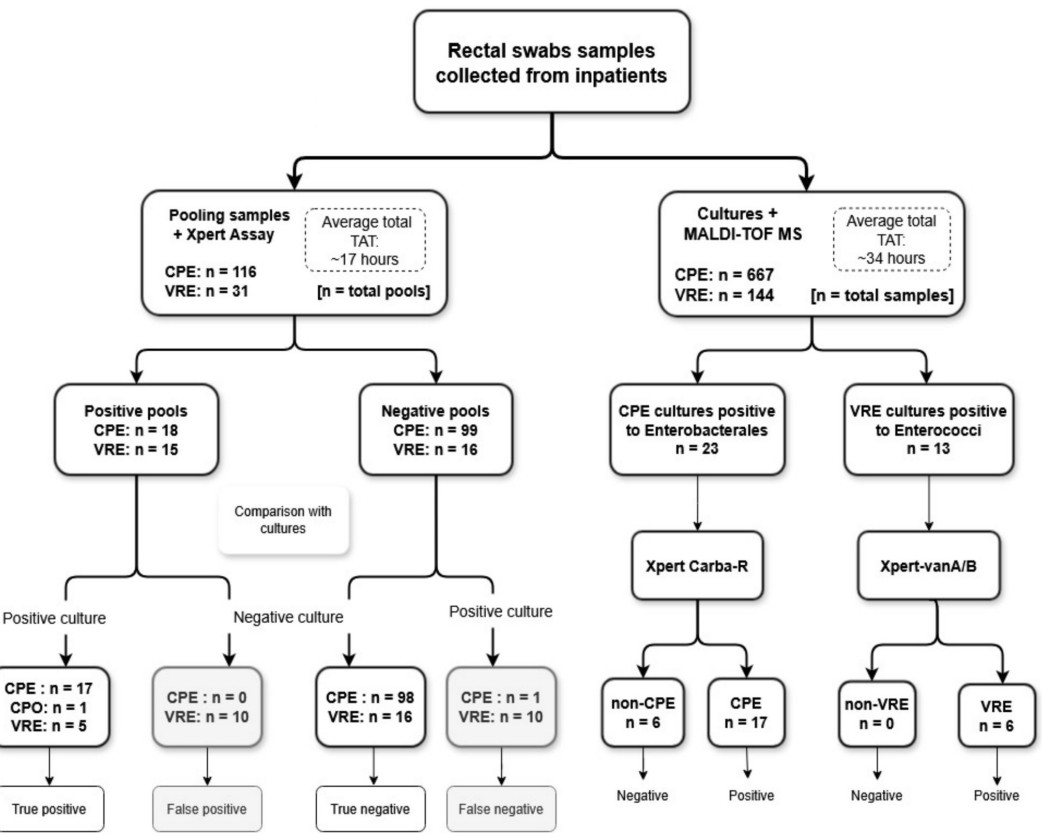

**FIG 1** Workflow for screening carbapenemase-producing *Enterobacterales* (CPE) and vancomycin-resistant enterococci (VRE) using molecular pooling techniques and comparative culture-based methods.

### Pooling protocol

Retrospective swab samples stored in the lab for up to 7 days were used for pooling by placing each swab in 1 mL saline and incubating for 5 minutes. From each tube, 100 µL was taken to a pool tube. Most pools consisted of six samples (total volume: 600 µL). From the pool tube, 200 µL was transferred to the sample reagent bottle provided in the Xpert Carba-R kit or Xpert *vanA/vanB* kit, vortexed thoroughly, and transferred into the cartridge, according to the manufacturer's instructions.

### Clinical phase

Swab samples were collected prospectively from inpatients and tested in two ways in parallel by PCR pooling and individual cultures. Each swab was first placed in 1 mL saline and incubated for 5 minutes. From the suspension, 10 µL was plated on chromogenic agar for culture, as described in the pre-clinical phase, and 100 µL was used for pooling. Saline-suspended swab samples from three to seven patients were pooled into one tube, with most pools containing six samples (140 pools, 95%). From each pool tube, 200 µL was transferred to the sample reagent bottle, vortexed thoroughly, and transferred into the cartridge, according to the manufacturer's instructions for testing using Xpert Carba-R PCR or Xpert *vanA/vanB* PCR (Cepheid, Sunnyvale, CA, USA). The reference method was defined as culture-based screening combined with PCR confirmation of colonies. During this phase, negative pooled PCR results were reported as negative for CPE/VRE in the laboratory information system. However, all samples were also individually cultured. If a positive culture result was obtained despite a negative PCR pool (i.e., a false negative), laboratory staff immediately notified the nursing team.

Since the presence of carbapenemases in *Pseudomonas aeruginosa* and *Acinetobacter baumannii* is of clinical importance, we did not consider their detection by the Xpert Carba-R as a false-positive result.

### Turnaround time

Total TAT consisted of two components: (i) pre-analytical time, defined as the interval from sample order to receipt in the laboratory, and (ii) analytical and post-analytical time, defined as the interval from laboratory receipt to result confirmation and reporting. These time intervals were calculated using data extracted from the laboratory information system.

### Statistical analysis

Statistical analyses and visualizations were performed in Python (version 3.11) using the following packages: numpy and pandas for data manipulation, seaborn and matplotlib for plotting, scipy.stats for non-parametric testing (Kruskal–Wallis), and scikit-posthocs for Bonferroni-adjusted Dunn's pairwise comparisons. A *P*-value < 0.05 was considered statistically significant.

## RESULTS

### CPE pooling

During the pre-clinical phase of the trial (validation), 115 CPE retrospective samples were divided into 19 pools containing three to seven samples per pool; of these, 11 pools contained a positive CPE sample and 8 did not. The pools were subjected to testing by Xpert PCR, and all eight negative pools were also negative by Xpert, while 10 of 11 positive pools tested positive by Xpert, and one tested negative. This test-negative pool contained an *E. coli* carrying New Delhi metallo-beta-lactamase-1 (NDM). One of the culture-negative PCR pools tested positive for NDM. This was traced to an NDM-producing *Acinetobacter baumannii*. We did not consider this a false-positive result.

The total accuracy of the CPE pooling test was 99.78%, with 94.44% (74.2–99.0 CI 95%) sensitivity and 100% (99.1–100) specificity. The positive predictive value (PPV) was

**TABLE 1** Performance values for Xpert Carba-R PCR using the pooling method during the clinical phase[a]

| Group | Accuracy (%) | Sensitivity (%) | Specificity (%) | PPV (%) | NPV (%) |
|---|---|---|---|---|---|
| CPE (n = 116) | 99.78 | 94.4 (74.2–99.0) | 100 (99.1–100) | 100 (81.6–100) | 99.78 (98.7–100) |

[a]Values in parentheses are the 95% CI.

100% (81.6–100), and the negative predictive value (NPV) was 99.78% (98.7–100). When considering each gene separately, OXA-48 and KPC were all detected correctly according to cultures, while one out of five NDM pools missed a positive sample (Table S1).

During the clinical phase, we collected rectal swabs from 667 patients for CPE screening. The CPE samples were collected into a total of 116 pools, 99 pools were negative in the PCR test, while the other 17 were positive (Table 1). There was one false-negative PCR pool that missed a *Citrobacter freundii* carrying the KPC gene, and one PCR pool that was positive for the NDM gene, which originated from an isolate of *Acinetobacter baumannii*; therefore, it was not included as a false positive. The total CPE performance was very high. No cases positive for IMP were detected during the study period (Table S2).

The mean threshold cycle level (Ct) values during the clinical phase of the study were as follows (Fig. 2; Table S3): 31.34 for *bla*OXA-48 (n = 6), 31 for *bla*KPC (n = 3), 28.34 for *bla*NDM (n = 7), and 30.1 for *bla*VIM (n = 1). There were no false positives for any of these genes.

## VRE pooling

During the pre-clinical phase of the study, 120 VRE retrospective samples were collected into 20 pools containing three to seven samples per pool. Of these, 1 was PCR negative, and 19 were positive. While 11 of the pools were in concordance with the result of the culture, 9 pools were not. The total VRE accuracy was 77.5%, with a specificity of 50% and sensitivity of 100%. PPV was 71%, and NPV was 100%. For *vanA*, accuracy, sensitivity, specificity, PPV, and NPV were all 100%. For *vanB*, accuracy and specificity were low (Table S4).

Following the results of the pre-clinical phase, the goal of the clinical phase of the study was to assess the performance of the PCR pooling for the detection of VRE-*vanA* colonization. During the clinical phase, rectal swabs from 144 patients were collected into a total of 31 pools. Of these, 16 were PCR negative, and 15 were positive. While 18 of the pools were in concordance with the culture result, in 13 pools, there was no growth on cultures, probably due to the detection of *vanB* genes from non-*Enterococcus* species. The performance of van-A was relatively good with 93.55% accuracy, 100% sensitivity (51–100 CI 95%), 92.59% specificity (76.6–98.7), and NPV of 100% (86.8–100); however,

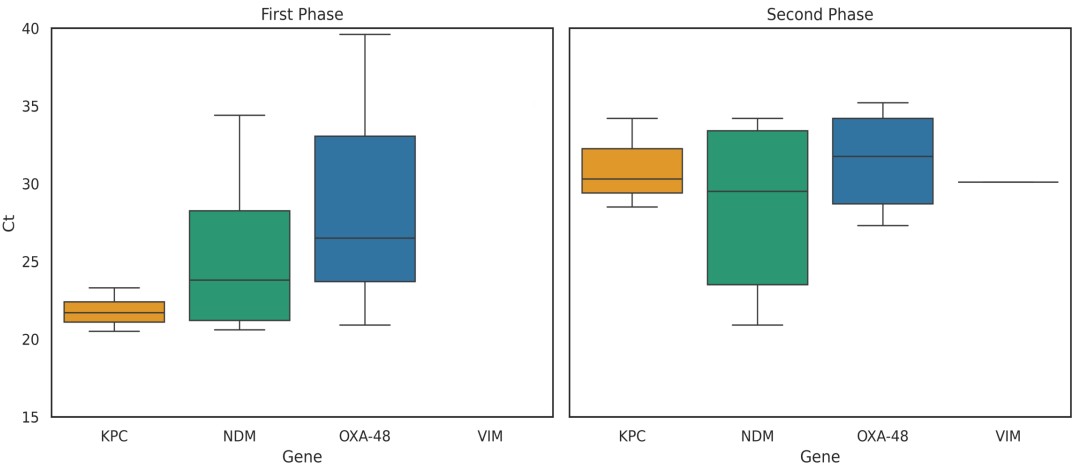

**FIG 2** Cycle threshold (Ct) value distribution of carbapenemase resistance genes detected by Xpert Carba-R PCR assay.

**TABLE 2** Performance values for Xpert *vanA/vanB* PCR using the pooling method, during the clinical phase[a]

| Group | Accuracy (%) | Sensitivity (%) | Specificity (%) | PPV (%) | NPV (%) |
|---|---|---|---|---|---|
| VRE (*n* = 31) | 77.42 | * | 75.8 (63.5–85.0) | * | 100 (92.0–100) |
| Van-A | 93.55 | 100 (51–100) | 92.59 (76.6–98.7) | 66.6 (30–94.1) | 100 (86.8–100) |
| Van-B | 61.29 | * | 61.29 (43.8–76.3) | * | 100 (83.2–100) |

[a]Values in parentheses are the 95% CI. Values marked by "*" could not be calculated.

it had a relatively low PPV (66.6%) due to two false-positive pools. For *vanB*, however, accuracy and specificity were low (61.3%) due to a high false-positive rate, and sensitivity and PPV could not be determined since there were no true-positive samples (Table 2; Table S5).

For *vanA*, the mean Ct values during the pre-clinical and clinical phases of the study were 21.2 (*n* = 17) and 21.9 (*n* = 4), respectively (Fig. 3). The mean Ct value for false positives was 31.9 (*n* = 2). For *vanB*, the mean Ct value for true positives during the pre-clinical phase of the study was 26.7 (*n* = 5). The mean Ct value for false positives was 34.5 (*n* = 9) in the pre-clinical phase and 32.1 (*n* = 12) during the clinical phase. These results indicate a higher likelihood of high Ct values to be false positives, for both *vanA* and *vanB*, which requires further investigation.

## Turnaround time

The clinical phase of the study was conducted in March 2024. To assess the impact of PCR pooling on TAT for CPE and VRE screening, we compared the total TAT of culture-only screening in February and April 2024 with the combined approach of PCR pooling and cultures used in March 2024. Since the clinical phase involved a comparison with cultures, the most significant reduction in TAT was observed for negative pools. Overall, the mean time from specimen collection to result (i.e., total TAT) for CPE screening decreased by 50%, from ~34 hours in February to ~17 hours in March (median time decreased from 35.01 ± 9.63 hours to 14.9 ± 11.9 hours. Kruskal $H$ = 827.97, $P < 0.001$). VRE screening showed a more modest decline of ~6 hours (mean and median) from February to March (Kruskal $H$ = 26.27, $P < 0.001$), largely because only 14% of VRE pools were truly negative and most required culture follow-up (Fig. 4 and Table 3). Although the Xpert test can be completed in less than an hour, the time from specimen collection to testing (i.e., pre-analytical) varies due to several factors: transportation time, workload, shift timing (morning/evening), the time required to gather all screening tests for pooling, and the time needed for analysis and reporting. These factors also vary between different institutes.

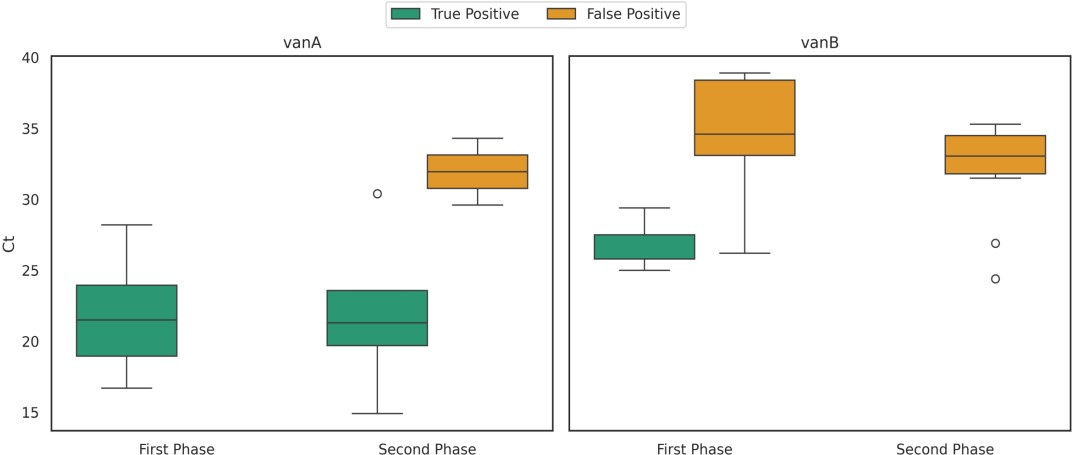

**FIG 3** Cycle threshold (Ct) value distribution of vancomycin resistance genes, *vanA* and *vanB*.

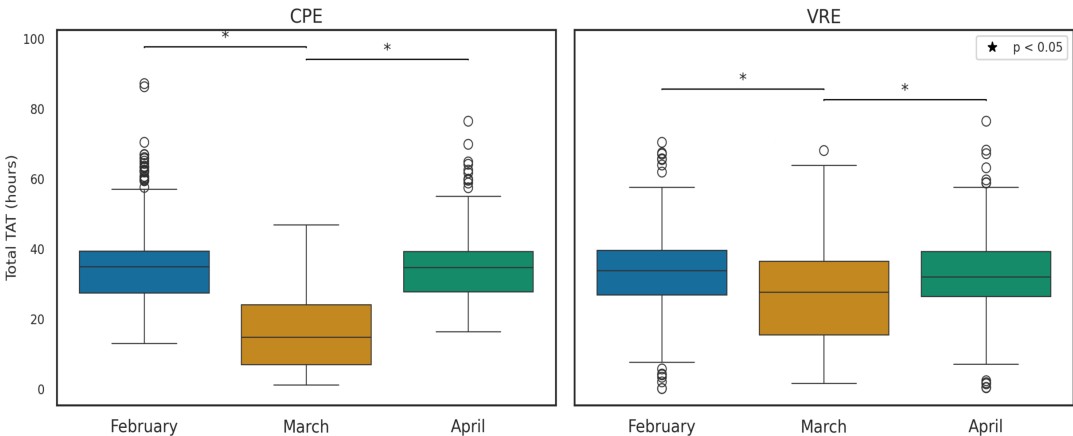

**FIG 4** Monthly comparison of laboratory turnaround times for CPE and VRE screening, showing average hours from collection to result during months of using cultures exclusively (February and April) compared to a month using PCR pooling alongside cultures (March).

We further compared monthly turnaround times across the individual workflow phases: pre-analytical (defined as the interval from test order—assumed equivalent to sample collection—to laboratory receipt) and analytical + post-analytical (from laboratory receipt to result confirmation). Due to limitations in measuring the time between test result and result confirmation and report, result confirmation was used as the endpoint of the analytical phase, effectively combining the analytical and post-analytical phases into a single interval. Statistical comparisons were performed using the Kruskal–Wallis test, followed by Dunn's pairwise comparisons with Bonferroni correction for multiple testing (Table S6).

For CPE, the pre-analytical TAT in April was modestly but significantly longer than in February and March (median + 0.6 hour and + 1.1 hours, respectively; $P < 0.01$ and $P < 0.001$), likely reflecting holiday-related transport delays. In contrast, the analytical TAT was dramatically shorter in March, the pooling month: February → March median $\Delta = -16.8$ hours ($P < 0.001$) and April → March $\Delta = -15.9$ hours ($P < 0.001$). These analytical-phase savings account for nearly all the reduction in total CPE TAT observed in March. For VRE, pre-analytical TATs did not differ significantly between months ($P \geq$

**TABLE 3** Monthly CPE and VRE turnaround time (hours) for each phase: summary statistics (median, mean, SD, and 95% CI)

| Test group | Month | N | Phase | Mean | Median | SD | CI 95% low (median) | CI 95% high (median) |
|---|---|---|---|---|---|---|---|---|
| CPE | February | 838 | Pre-analytical | 5.21 | 2.98 | 5.07 | 2.57 | 3.55 |
| | | | Analytical | 29.38 | 25.69 | 8.84 | 25.40 | 26.04 |
| | | | Total | 34.59 | 35.01 | 9.63 | 33.96 | 35.50 |
| | March | 788 | Pre-analytical | 4.74 | 2.57 | 4.72 | 2.18 | 3.08 |
| | | | Analytical | 12.57 | 6.43 | 11.53 | 5.83 | 7.43 |
| | | | Total | 17.32 | 14.90 | 11.91 | 13.95 | 15.83 |
| | April | 817 | Pre-analytical | 5.81 | 4.20 | 4.94 | 3.68 | 4.88 |
| | | | Analytical | 28.48 | 25.03 | 8.95 | 24.80 | 25.52 |
| | | | Total | 34.29 | 34.80 | 9.04 | 34.30 | 35.22 |
| VRE | February | 196 | Pre-analytical | 4.98 | 2.78 | 4.69 | 2.18 | 4.03 |
| | | | Analytical | 28.53 | 25.53 | 10.87 | 24.96 | 26.52 |
| | | | Total | 33.51 | 33.88 | 11.60 | 32.35 | 35.28 |
| | March | 179 | Pre-analytical | 5.05 | 4.03 | 4.46 | 2.62 | 4.93 |
| | | | Analytical | 22.11 | 23.90 | 13.73 | 22.37 | 24.37 |
| | | | Total | 27.16 | 27.77 | 14.08 | 24.60 | 31.08 |
| | April | 185 | Pre-analytical | 4.93 | 3.23 | 4.47 | 2.43 | 4.48 |
| | | | Analytical | 28.35 | 25.50 | 11.09 | 24.52 | 26.13 |
| | | | Total | 33.28 | 32.03 | 11.28 | 31.12 | 34.33 |

**TABLE 4** Comparison of Xpert Carba-R performance across different studies

| Study | Sample type | No. of specimens | Genes | Sensitivity (%) | Specificity (%) | PPV (%) | NPV (%) |
|---|---|---|---|---|---|---|---|
| Jin et al. (15) (multicenter evaluation) | Rectal swabs | 2,404 | All | 96.0 | 94.0 | 69.5 | 99.4 |
| | | | $bla_{KPC}$ | 94.5 | 99.6 | 95.4 | 99.5 |
| | | | $bla_{NDM}$ | 96.0 | 97.7 | 69.6 | 99.8 |
| | | | $bla_{IMP}$ | 100 | 94.7 | 11.3 | 100 |
| | | | $bla_{VIM}$ | 100 | 99.9 | 50 | 100 |
| | | | $bla_{OXA-48}$ | 100 | 99.9 | 40 | 100 |
| | Isolates | 2,404 | All | 99.7 | 98.0 | 98.8 | 99.5 |
| | | | $bla_{KPC}$ | 99.5 | 98.8 | 97.9 | 99.7 |
| | | | $bla_{NDM}$ | 100 | 99.8 | 99.1 | 100 |
| | | | $bla_{IMP}$ | 100 | 99.8 | 95.0 | 100 |
| | | | $bla_{VIM}$ | 100 | 100 | 100 | 100 |
| | | | $bla_{OXA-48}$ | 100 | 99.8 | 90 | 100 |
| Moubareck et al. (16) | Rectal swabs | 1,813 | All | 80.3 | 99.8 | NA | NA |
| Cury et al. (17) | Rectal swabs | 921 | All | 94.0 | 98.6 | 86.8 | 99.4 |
| Ko et al. (18) | Rectal swabs | 408 | All | 100 | 96.7 | 53.6 | 100 |
| Traczewski et al. (19) | Rectal swabs + isolates | 633 | All | 97 | 98.6 | 95.3 | 99 |
| | | | $bla_{KPC}$ | 96.7 | 99.3 | 87.9 | 100 |
| | | | $bla_{NDM}$ | 100 | 99.8 | 96.3 | 100 |
| | | | $bla_{IMP}$ | 96.3 | 100 | 99.8 | 99.8 |
| | | | $bla_{VIM}$ | 93.5 | 99.8 | 96.7 | 100 |
| | | | $bla_{OXA-48}$ | 95 | 99.8 | 97.4 | 100 |
| Hoyos-Mallecot et al. (20) | Rectal swabs | 241 | $bla_{KPC}$, $bla_{OXA-48}$, $bla_{NDM}$ | 100 | 99.1 | 85.7 | 100 |
| Tato et al. (21) | Rectal swabs | 383 | All | 96.6 | 98.6 | 95.3 | 99.0 |
| Smith et al. (22) | CPE Isolates | 129 | All | 100 | 100 | 100 | 100 |
| Moore et al. (23) | Rectal swabs | 755 | All | 97.7 | 97.2 | 95.8 | 98.4 |

0.10). However, the analytical phase showed a significant reduction in March (February → March median Δ ≈ −6.4 hours, $P < 0.001$), explaining the smaller decrease in overall VRE TAT (Table S6).

## Literature review of Xpert Carba-R and Xpert *vanA/B* performance

We summarized the main studies that evaluated the performance of Xpert Carba-R assay (Table 4) using rectal swabs or carbapenem-producing organism (CPO) isolates. The mean performance values for all studies that used rectal swabs (considering equal weight for each study) for CPE screening were as follows: 95.2% sensitivity, 97.8% specificity, 83.1% PPV, and 99.3% NPV. In studies that have evaluated the performance of the assay for isolates separately, the values were higher.

**TABLE 5** Comparison of Xpert *vanA/vanB* performance across different studies

| Study | Sample type | No. of specimens | Genes | Sensitivity (%) | Specificity (%) | PPV (%) | NPV (%) |
|---|---|---|---|---|---|---|---|
| Babady et al. (24) | Rectal swabs | 300 | *vanA* | 100 | 96.9 | 91.3 | 100 |
| Holzknecht et al. (25) | Rectal swabs | 1,110 | *vanA* | 87.1 | 99.7 | 98.0 | 97.7 |
| | | | *vanB* | 77.6 | NA[a] | 0.4 | NA |
| Zabicka et al. (26) | Stool samples | 37 | *vanA*, *vanB* | 61.5 | 79.2 | 61.5 | 79.2 |
| Bourdon et al. (27) | Rectal swabs | 804 | *vanA*, *vanB* | 100 | 85.4 | 8.7 | 100 |
| | | | *vanA* | 100 | 99.5 | 66.7 | 100 |
| | | | *vanB* | 100 | 85.6 | 2.6 | 100 |
| Gazin et al. (28) | Stool samples | 23 | *vanA* | 73.9 | 92.6 | 89.5 | 80.6 |
| | | 16 | *vanB* | 87.5 | 14.7 | 32.6 | 71.4 |
| Dekeyser et al. (29) | NA | 118 | *vanA*, *vanB* | 100 | 76.8 | 2.8 | 100 |
| | | 447 | *vanA*, *vanB* | 100 | 69.3 | 15.0 | 100 |

[a]NA, not available.

We also summarized the results of studies that evaluated the performance of Xpert *vanA/vanB* (Table 5) using rectal swabs or stool samples. These data demonstrate that while the Xpert *vanA/vanB* assay has excellent sensitivity, the specificity and positive predictive value (especially for *vanB*) show considerable variation across studies.

## DISCUSSION

Active surveillance of carbapenem-resistant *Enterobacterales* and vancomycin-resistant *Enterococcus* colonization among hospitalized patients has become a routine procedure in healthcare settings across the world. Given the limited treatment options in cases of infections caused by CPE and VRE, the focus remains on infection control measures. Thus, rapid detection of CPE/VRE colonization has become a critical need (4, 7, 30). Carbapenem-resistant *Enterobacterales* remain on the WHO critical list of bacterial pathogens for 2024 due to the need for research and development of new therapies, their wide resistance and increased prevalence (31).

The first implementation of the sample pooling technique was reported by Dorfman in 1943 (32). The use of pooling for PCR assays began in the early 2000s (33) and gained significant attention during the COVID-19 pandemic, with large-scale implementation of pooled RT-PCR reported by several studies that highlighted its efficiency in utilizing the huge demand of PCR testing (13, 14). The pooling strategy has shown many advantages regarding testing capacity and costs, as well as its utilization for large-scale population surveillance and screening of asymptomatic SARS-CoV-2 carriers as part of active measures for the reduction of transmission events (34). The pooling method should consider the prevalence of the examined disease/genotype in the population, as well as the performance of the specific assay, when determining the number of samples that can be pooled and tested together. For a low prevalence of the disease and good assay performance, the pooling method can be highly efficient in terms of turnaround time, costs, and work overload. The efficiency of the pooling method declines when prevalence reaches 30%, and beyond that, individual testing is more appropriate (34).

We present an evaluation study for a PCR pooling technique for the detection of carbapenemase genes or vancomycin resistance genes from rectal swab samples. In the pre-clinical phase of the study, we have used retrospective samples with known results according to the culture-based method. In the clinical phase of the study, we tested new samples using the PCR pooling technique and compared them with culture-based methods.

The CPE screening using PCR pooling demonstrated high performance, with a sensitivity of 94.4%, specificity of 100%, and total accuracy of >99%. However, the small number of individual CPE-positive samples limited the ability to conduct subgroup analyses. For instance, a lower sensitivity for *bla*KPC was observed during the clinical phase, though this was attributed to a single false-negative result in one pooled sample. This discrepancy may have been influenced by the use of retrospective specimens, collected up to 7 days prior to testing. It is important to note that, according to the Xpert Carba-R IFU, storage of swab samples in collection tubes should be at 15°C–28°C for up to 5 days. Thus, the use of retrospective samples stored for a longer time could result in lower sensitivity. In addition, pooling can impact the assay's limit of detection, potentially leading to false negatives due to sample dilution, as detection sensitivity depends on bacterial concentration (35). Regarding *bla*VIM, although the test performed well, only one sample was positive, limiting the ability to accurately assess sensitivity and positive predictive value. No *bla*IMP-positive samples were detected in this study, consistent with its low prevalence in our region. Therefore, additional testing with confirmed *bla*IMP-positive samples is necessary to fully evaluate the method's performance for this target. One PCR pool was positive for *bla*NDM, while culture results did not identify any CPE growth but rather *Acinetobacter baumannii* carrying the gene. While *A. baumannii* is non-CRE, it is considered a CPO, is listed among the multidrug-resistant agents that are commonly screened for, and its presence requires contact transmission-based precautions, similar to those performed for CPE (36).

According to our summary of the main studies that tested the performance of Xpert Carba-R, the mean performance values for all studies that used rectal swabs (considering equal weight for each study) were as follows: 95.2% sensitivity, 97.8% specificity, 83.1% PPV, and 99.3% NPV. For studies that have evaluated the performance of the assay for isolates separately, the values were higher, approaching 100%. A previous study that tested the pooling strategy of CPE rectal swabs using Xpert Carba-R also reported very high performance rates of 100% sensitivity, 99.0% specificity, 88.2% PPV, and 100% NPV, for pools of five samples and a positivity rate of 5.3% (37). Our results from the pooling PCR assay showed overall good performance values, similarly to those of the studies mentioned above.

VRE screening using PCR pooling demonstrated good performance for *vanA*, showing high sensitivity and specificity. However, the PPV value was reduced due to a small number of false-positive pools. For *vanB*, sensitivity could not be determined during the clinical phase, as only false-positive pools were detected. These findings align with previous studies that showed very good performance of *vanA* and lower specificity of *vanB* with very low PPV (25, 38, 39). The high rate of *vanB* false positives is thought to result from the presence of anaerobic gut bacteria, such as *Clostridium* and *Ruminococcus* species (40), suggesting this is not a technical issue but rather a clinical diagnostic matter that should be taken into account. To overcome the high rate of *vanB* false positives (non-*Enterococcus vanB* genes), the *vanB* test should be confirmed with cultures. We observed that the mean Ct values of true-positive pools for both *vanA* and *vanB* were lower than the mean Ct values of false positives, suggesting that Ct thresholds may help distinguish between true and false positives. However, this trend must be further evaluated using larger data sets, with comparisons to individual swab Ct values and varying pool sizes to determine an appropriate Ct cutoff and the true limit of detection. Although VRE screening remains highly essential—especially in immunocompromised patients due to the risk of nosocomial outbreaks and bloodstream infections with high mortality rates (1), current methods still require culture confirmation to ensure result accuracy. Several studies have reported notable percentages of false-positive results for *vanB* (41), often due to the detection of intrinsic *vanB* genes in non-enterococcus species that lack clinical significance compared to plasmid-borne *vanB* (40). Such false-positive results, if not confirmed by cultures as true VRE, can lead to unnecessary isolation of patients, burden clinical care, staff resources, and hospital costs (41). Given the frequent detection of *vanB* in non-VRE organisms, we believe it is worth reconsidering whether *vanB* should be routinely included in PCR-based VRE screening assays.

Several limitations of the pooling technique should be considered: (i) in cases of low microbial load, the pooling can lead to reduced concentration and fall below the assay's limit of detection. (ii) Different specimen types can have different effects on the pooling performance due to the combination of potential PCR inhibitors. Thus, each pooling protocol should be validated according to specimen type. (iii) The limit of detection should be determined for each gene separately, and subsequently, the interpretation of results (post-analytical phase) should consider it (13, 34).

One of the limitations of our study is that we could not retrieve the Ct value of direct PCR from individual tests, and these were only kept as "detected"/"not detected"; therefore, we were unable to determine the limit of detection. Further studies should compare Ct values from individual swabs and pooled samples using varying pool sizes to identify an optimal Ct cutoff and determine both the detection limit and the most appropriate pooling strategy.

The demand for CPE/VRE screening in hospital settings is increasing, yet culture-based methods remain labor-intensive and time-consuming. The implementation of PCR pooling can enable both large-scale screening and a reduction of turnaround time. In our clinical study, the average TAT was reduced by 50%, from approximately 34 hours to ~17 hours. Although the test run time is less than an hour, the comparison with cultures extended the time for the result. Adopting PCR pooling as the primary screening method could further reduce TAT to several hours only. This substantial decrease has important

implications for infection prevention, allowing for earlier identification and isolation of colonized patients, thereby improving outbreak control and patient management.

While PCR assays initially carry a higher cost compared to traditional culture methods, numerous studies have demonstrated the cost-effectiveness of rapid CRE screening using PCR methods. This approach has been shown to significantly reduce patient length of stay, ultimately lowering overall per-patient costs (42–45). The economic benefits of PCR-based screening stem from its ability to provide faster results, enabling faster implementation of appropriate infection control measures and targeted treatment strategies.

In our institute, the positivity rate of CPE and VRE is 1.4% and 4.7%, respectively, according to screening tests of 2024. This, together with the performance results and potential cost-effectiveness, suggests that the pooling method can be highly efficient for the active surveillance of CPE colonization in healthcare settings when the population shows low prevalence.

## Limitations

Limitations of this study include the generally low number of individual CPE genes, with especially low numbers of positive genes of *bla*IMP and *bla*VIM. In addition, in the clinical phase of the trial, no enterococci carrying *vanB* were detected.

## ACKNOWLEDGMENTS

We would like to express our gratitude to Dr. Christophe Martinaud for his thoughtful suggestions, which greatly improved the manuscript.

## AUTHOR AFFILIATIONS

[1]Emerging Infectious Diseases Research Laboratory, Samson Assuta Ashdod University Hospital, Ashdod, Israel

[2]Clinical Microbiology Laboratory, Samson Assuta Ashdod University Hospital, Ashdod, Israel

[3]Infectious Diseases Unit, Samson Assuta Ashdod University Hospital, Ashdod, Israel

[4]Faculty of Health Sciences, Ben Gurion University in the Negev, Beer Sheba, Israel

[5]National Institute for Antibiotic Resistance and Infection Control, Tel Aviv, Israel

[6]Faculty of Medical and Health Sciences, Tel Aviv University, Tel Aviv, Israel

[7]Medison Pharma, Petah Tikva, Israel

## AUTHOR ORCIDs

Danielle Keidar-Friedman http://orcid.org/0000-0002-2028-0850

## AUTHOR CONTRIBUTIONS

Danielle Keidar-Friedman, Formal analysis, Investigation, Methodology, Software, Validation, Visualization, Writing – original draft, Writing – review and editing | Larissa Gil, Investigation, Methodology, Project administration, Writing – review and editing | Anka Tsur, Conceptualization, Investigation, Methodology, Project administration, Writing – review and editing | Tal Brosh-Nissimov, Conceptualization, Writing – review and editing | Yehuda Carmeli, Conceptualization, Writing – original draft, Writing – review and editing | Boaz David Rosenfeld, Conceptualization, Project administration, Resources, Writing – review and editing | Nadav Sorek, Conceptualization, Funding acquisition, Methodology, Project administration, Resources, Writing – original draft, Writing – review and editing

## ADDITIONAL FILES

The following material is available online.

## Supplemental Material

**Supplemental material (Spectrum01080-25-s0001.xlsx).** Tables S1 to S6 and raw data.

## Open Peer Review

**PEER REVIEW HISTORY (review-history.pdf).** An accounting of the reviewer comments and feedback.

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
