## [Reviewer comments · Microbiology Spectrum]

Microbiology Spectrum

Evaluation of sample pooling using Xpert® Carba-R & Xpert® vanA/vanB PCR for screening of carbapenemase-producing Enterobacterales (CPE) and vancomycin-resistant Enterococci (VRE) colonization

Danielle Keidar-Friedman, Larissa Gil, Anka Tsur, Tal Brosh-Nissimov, Yehuda Carmeli, Boaz Rosenfeld, and Nadav Sorek

Corresponding Author(s): Danielle Keidar-Friedman, Assuta Ashdod Hospital

Review Timeline:

Submission Date:	April 8, 2025
Editorial Decision:	June 11, 2025
Revision Received:	June 25, 2025
Accepted:	July 10, 2025

Editor: Siu-Kei Chow

Reviewer(s): Disclosure of reviewer identity is with reference to reviewer comments included in decision letter(s). The following individuals involved in review of your submission have agreed to reveal their identity: Masahiro Suzuki (Reviewer #1); Ya Wang (Reviewer #2)

Transaction Report:

DOI: <https://doi.org/10.1128/spectrum.01080-25>

Re: Spectrum01080-25 (Evaluation of sample pooling using Xpert® Carba-R & Xpert® vanA/vanB PCR for screening of carbapenemase-producing Enterobacterales (CPE) and vancomycin-resistant Enterococci (VRE) colonization)

Dear Dr. Danielle Keidar-Friedman:

Thank you for the privilege of reviewing your work. Below you will find my comments, instructions from the Spectrum editorial office, and the reviewer comments.

Revision Guidelines

Sincerely,
Siu-Kei Chow
Editor
Microbiology Spectrum

Reviewer #1 (Comments for the Author):

The concept of PCR screening using pooled samples is valuable for improving turnaround time (TAT) and economic efficiency. This study has potential to contribute to the implementation of pooled PCR screening in clinical settings. However, the methodological criteria and supporting data are insufficient to recommend this study as a reference for evaluating pooled PCR strategies.

Methods

Pre-clinical Phase

The manuscript states that bacteria were isolated and carbapenemase genes were detected using both swab samples and bacterial isolates. However, the pooling protocol is not described in the Methods section. It is unclear whether pooling was conducted using bacterial isolates or swab samples. This information is critical, and the pooled PCR protocol for the pre-clinical phase must be clearly and precisely described.

Clinical Phase

The authors mention two approaches for bacterial culture: one using the same method as the pre-clinical phase and another using saline-suspended samples. Please clarify this point. If this interpretation is correct, indicate which set of results served as the reference or "gold standard" for performance evaluation.

Turnaround Time (TAT)

The method for calculating TAT is not described. The manuscript should provide the raw data used to calculate TAT, as well as a clear explanation of the calculation method.

Results

Examination of the CRE raw data reveals that several Ct values in the clinical phase were high, approaching the detection limit. To assess the impact of pooling, it is essential to compare Ct values from individual positive swab samples with those from the corresponding pooled samples. These comparisons are necessary to evaluate the detection limit and determine an appropriate pooling size. Please provide Ct values for single positive swab samples in the clinical phase. If swab samples were also used in the pre-clinical phase, corresponding Ct values should be included as well.

In the VRE raw data, pooled samples 990910018 and 990910014 show a significant increase in Ct values for vanA in the presence of vanB-positive samples, suggesting interference. However, in sample 990946002, the Ct value for vanA was low despite the presence of vanB. The vanB-positive result in sample 990946002 is questionable. Was the vanB strain confirmed using an alternative primer set or method?

Discussion

Lines 282-283: The statement that "the pooling technique does not have a negative effect on the performance of the PCR assay" appears to be an overstatement. The sensitivity of PCR is influenced by the input volume and sample dilution. Without providing Ct values from single positive samples, it is unclear how pooling affected detection sensitivity. The authors should discuss the limitations of pooling by comparing Ct values between single and pooled samples.

Lines 288-290: If the false positives for vanB are attributed to anaerobic bacteria, why were similar false positives observed in the pre-clinical phase if isolates were used for pooling? This discrepancy should be addressed.

Reviewer #2 (Comments for the Author):

In this study, the authors compared a PCR pooling strategy against standard culture-based methods for detecting carbapenemase-producing Enterobacteriales (CPE) and vancomycin-resistant Enterococci (VRE) colonization. It is a practical study trying to address the growing need for cost-effective and rapid screening methods for CPE in hospital settings. I think the study is overall well-conceived and potentially impactful, but it would benefit from addressing the following points:

1. Address impact of retrospective sample handling:

The lower sensitivity observed in the clinical phase for blaKPC, though based on a single false-negative result, was partially attributed to the age of retrospective samples. The authors should discuss more clearly how delays between sample collection and testing might affect bacterial DNA integrity or detection thresholds in pooled PCR. If possible, including a brief sensitivity analysis or proposing specific guidelines for acceptable storage durations would improve the robustness of the findings.

2. Enhance workflow detail in Fig1:

While the reduction in turnaround time (TAT) is clearly demonstrated, the workflow in Figure 1 could be more informative. I recommend integrating practical implementation details, such as the estimated TAT for each step, reagent or equipment costs, and manpower requirements for both methods. This would significantly strengthen the translational value of the study for infection control teams considering adoption.

3. Clarify the clinical implications of vanB:

The false-positive rate for vanB and the absence of true vanB-positive Enterococci is clinically meaningful. The authors could discuss the significance of vanB being continually included in routine PCR panels for VRE screening, particularly in low-prevalence settings. Clarifying the clinical risks of missing true vanB cases versus the burden of false positives would help guide diagnostic policy.

4. Provide Ct value distributions for true vs. false positives:

The manuscript notes that true-positive results had lower Ct values than false positives, which is an important finding. Including actual Ct value ranges or a comparative figure (e.g., boxplot or density plot) would help support the idea that Ct thresholds may be useful for refining test interpretation. If not feasible in this study, this could be recommended for future work.

Dear Dr. Danielle Keidar-Friedman:

Thank you for the privilege of reviewing your work. Below you will find my comments, instructions from the Spectrum editorial office, and the reviewer comments.

Revision Guidelines

- Upload point-by-point responses to the issues raised by the reviewers in a file named "Response to Reviewers," NOT in your cover letter.
- Upload a compare copy of the manuscript (without figures) as a "Marked-Up Manuscript" file.
- Upload a clean .DOC/.DOCX version of the revised manuscript and remove the previous version.
- Each figure must be uploaded as a separate, editable, high-resolution file (TIFF or EPS preferred), and any multipanel figures must be assembled into one file.
- Any [supplemental material](https://journals.asm.org/writing-your-paper#supplemental-material) intended for posting by ASM should be uploaded with their legends separate from the main manuscript. You can combine all supplemental material into one file (preferred) or split it into a maximum of 10 files with all associated legends included.

For complete guidelines on revision requirements, see our [Submission and Review Process](https://journals.asm.org/journal/spectrum/submission-review-process) webpage. Submission of a paper that does not conform to guidelines may delay acceptance of your manuscript.

Data availability: ASM policy requires that data be available to the public

upon online posting of the article, so please verify all links to sequence records, if present, and make sure that each number retrieves the full record of the data. If a new accession number is not linked or a link is broken, provide Spectrum production staff with the correct URL for the record. If the accession numbers for new data are not publicly accessible before the expected online posting of the article, publication may be delayed; please contact production staff (Spectrum@asmusa.org) immediately with the expected release date.

Publication Fees: For information on publication fees and which article types are subject to charges, visit our [website](https://journals.asm.org/publication-fees). If your manuscript is accepted for publication and any fees apply, you will be contacted separately about payment during the production process; please follow the instructions in that e-mail. Arrangements for payment must be made before your article is published.

ASM Membership: Corresponding authors may [join or renew ASM membership](https://www.asm.org/membership) to obtain discounts on publication fees. Need to upgrade your membership level? Please contact Customer Service at Service@asmusa.org.

The ASM Journals program strives for constant improvement in our submission and publication process. Please tell us how we can improve your experience by taking this quick [Author Survey](https://www.surveymonkey.com/r/ASMJournalAuthors).

Sincerely,
Siu-Kei Chow
Editor
Microbiology Spectrum

Reviewer #1 (Comments for the Author):

The concept of PCR screening using pooled samples is valuable for improving turnaround time (TAT) and economic efficiency. This study has potential to contribute to the implementation of pooled PCR screening in clinical settings.

However, the methodological criteria and supporting data are insufficient to recommend this study as a reference for evaluating pooled PCR strategies.

Thank you so much for your critical review of our manuscript.

Methods

Pre-clinical Phase

The manuscript states that bacteria were isolated and carbapenemase genes were detected using both swab samples and bacterial isolates. However, the pooling protocol is not described in the Methods section. It is unclear whether pooling was conducted using bacterial isolates or swab samples. This information is critical, and the pooled PCR protocol for the pre-clinical phase must be clearly and precisely described.

Pooling was conducted only using swab samples, not bacterial isolates.

We have added the detailed pooling protocol at the end of the pre-clinical Phase paragraph in the Methods section (lines 132-137).

Clinical Phase

The authors mention two approaches for bacterial culture: one using the same method as the pre-clinical phase and another using saline-suspended samples. Please clarify this point. If this interpretation is correct, indicate which set of results served as the reference or "gold standard" for performance evaluation.

During the clinical phase, we have used saline-suspended swab samples for the pooling PCR and for bacterial cultures. The bacterial cultures and Xpert PCR from colonies were used as the reference method. We have added this information to the methods section, with detailed explanation of the study design and the protocol of each phase of our study.

Turnaround Time (TAT)

The method for calculating TAT is not described. The manuscript should provide the raw data used to calculate TAT, as well as a clear explanation of the calculation method.

Total TAT was composed from two components: 1. Pre-analytical time, defined as the interval from sample order to receipt in the laboratory. 2. Analytical and post-analytical time, defined as the interval from laboratory receipt to result confirmation and reporting. These time intervals were calculated using data extracted from the laboratory information system.

We have generated a new figure (replacing **Figure 4**) that presents the distribution of the total TAT for each month and added a new table (**Table 3**) that shows the descriptive statistics of total TAT, as well as **Table S6** that shows the

statistical comparisons of TAT per each step (pre-analytical, analytical and total TAT) between the months. The raw data was added to the supplementary tables file. We have added a short paragraph with explanation of TAT calculation also in the methods (lines 156-161), and expanded the TAT results section (lines 237-270).

Results

Examination of the CRE raw data reveals that several Ct values in the clinical phase were high, approaching the detection limit. To assess the impact of pooling, it is essential to compare Ct values from individual positive swab samples with those from the corresponding pooled samples. These comparisons are necessary to evaluate the detection limit and determine an appropriate pooling size. Please provide Ct values for single positive swab samples in the clinical phase. If swab samples were also used in the pre-clinical phase, corresponding Ct values should be included as well.

Unfortunately, due to technical issues, we could not retrieve all the Ct values during the pooling month, and these were archived only as detected/not-detected results on our LIS (laboratory information system). The Ct values of the pools were handwritten by our staff, so we were able to retrieve those.

We believe the limit of detection should be tested in a follow-up study as you suggested, using a larger sample size. We have added this issue to the discussion.

In the VRE raw data, pooled samples 990910018 and 990910014 show a significant increase in Ct values for vanA in the presence of vanB-positive samples, suggesting interference. However, in sample 990946002, the Ct value for vanA was low despite the presence of vanB. The vanB-positive result in sample 990946002 is questionable. Was the vanB strain confirmed using an alternative primer set or method?

The presence of vanB was confirmed by culture and PCR from the pure culture of the *Enterococcus faecium* cultured from this swab sample.

Discussion

Lines 282-283: The statement that "the pooling technique does not have a negative effect on the performance of the PCR assay" appears to be an overstatement. The sensitivity of PCR is influenced by the input volume and sample dilution. Without providing Ct values from single positive samples, it is unclear how pooling affected detection sensitivity. The authors should discuss the

limitations of pooling by comparing Ct values between single and pooled samples.

We have made changes to the discussion about the limitation of our study, regarding the Ct values of direct PCR from individual tests, as well as general limitations of the pooling technique (lines 373-384).

Lines 288-290: If the false positives for vanB are attributed to anaerobic bacteria, why were similar false positives observed in the pre-clinical phase if isolates were used for pooling? This discrepancy should be addressed.

Swab samples were used for pooling in both phases. In the pre-clinical phase, we have used mostly positive samples in which *Enterococcus faecium* was detected, harboring vanA, vanB or both genes. In 9 pools of swab samples, although the Xpert PCR from colonies (per sample) did not detect vanB, it was detected in the pooled swab PCR – therefore, we considered it false positive.

Reviewer #2 (Comments for the Author):

In this study, the authors compared a PCR pooling strategy against standard culture-based methods for detecting carbapenemase-producing Enterobacterales (CPE) and vancomycin-resistant Enterococci (VRE) colonization. It is a practical study trying to address the growing need for cost-effective and rapid screening methods for CPE in hospital settings. I think the study is overall well-conceived and potentially impactful, but it would benefit from addressing the following points:

Thank you so much for your critical review of our manuscript.

1. Address impact of retrospective sample handling:

The lower sensitivity observed in the clinical phase for blaKPC, though based on a single false-negative result, was partially attributed to the age of retrospective samples. The authors should discuss more clearly how delays between sample collection and testing might affect bacterial DNA integrity or detection thresholds in pooled PCR. If possible, including a brief sensitivity analysis or proposing specific guidelines for acceptable storage durations would improve the robustness of the findings.

According to Cepheid Xpert Carba-R package insert, sample storage of swabs in transport tube should be 15–28 °C for up to 5 days for¹. We have added this

¹ Cepheid, “Xpert® Carba-R Assay (Package Insert) (Rev. B).”

information to our discussion (lines 256-258). Also, we have added a paragraph to the discussion about the limitations of the pooling method (lines 373-384).

2. Enhance workflow detail in Fig1:

While the reduction in turnaround time (TAT) is clearly demonstrated, the workflow in Figure 1 could be more informative. I recommend integrating practical implementation details, such as the estimated TAT for each step, reagent or equipment costs, and manpower requirements for both methods. This would significantly strengthen the translational value of the study for infection control teams considering adoption.

We have added more information to this figure, including the mean overall TAT of each phase and the sample size we had in each step, and expanded the results section (lines 237-270).

We could not include the reagent, equipment and manpower requirements since these were very complicate to calculate retrospectively. In addition, the costs might vary widely depending on the lab capabilities, therefore we think it might not be informative enough.

3. Clarify the clinical implications of vanB:

The false-positive rate for vanB and the absence of true vanB-positive Enterococci is clinical meaningful. The authors could discuss the significance of vanB being continually included in routine PCR panels for VRE screening, particularly in low-prevalence settings. Clarifying the clinical risks of missing true vanB cases versus the burden of false positives would help guide diagnostic policy.

We have added more to the discussion about vanB in the discussion section) (lines 345-372).

4. Provide Ct value distributions for true vs. false positives:

The manuscript notes that true-positive results had lower Ct values than false positives, which is an important finding. Including actual Ct value ranges or a comparative figure (e.g., boxplot or density plot) would help support the idea that Ct thresholds may be useful for refining test interpretation. If not feasible in this study, this could be recommended for future work.

We included the Ct values of pools for each gene, using boxplots and added Table S3 with the descriptive statistics for each gene.

For CPE we had only true positives, and for vanA/B we added both true and false

positives. Our data can only indicate a certain trend, but more experiments and larger sample size are needed to confirm this hypothesis.

Re: Spectrum01080-25R1 (Evaluation of sample pooling using Xpert® Carba-R & Xpert® vanA/vanB PCR for screening of carbapenemase-producing Enterobacterales (CPE) and vancomycin-resistant Enterococci (VRE) colonization)

Dear Dr. Danielle Keidar-Friedman:

Your manuscript has been accepted, and I am forwarding it to the ASM production staff for publication. Your paper will first be checked to make sure all elements meet the technical requirements. ASM staff will contact you if anything needs to be revised before copyediting and production can begin. Otherwise, you will be notified when your proofs are ready to be viewed.

Sincerely,
Siu-Kei Chow
Editor
Microbiology Spectrum